# Comparative Analysis of the GNAI Family Genes in Glioblastoma through Transcriptomics and Single-Cell Technologies

**DOI:** 10.3390/cancers15205112

**Published:** 2023-10-23

**Authors:** Ahmad Raza, Meng-Chi Yen, Gangga Anuraga, Iram Shahzadi, Muhammad Waqar Mazhar, Hoang Dang Khoa Ta, Do Thi Minh Xuan, Sanskriti Dey, Sachin Kumar, Adrian Wangsawijaya Santoso, Bianca Tobias William, Chih-Yang Wang

**Affiliations:** 1Graduate Institute of Cancer Biology and Drug Discovery, College of Medical Science and Technology, Taipei Medical University, Taipei 11031, Taiwan; ahmadrazashahid@gmail.com (A.R.); m654110001@tmu.edu.tw (D.T.M.X.); m654111005@tmu.edu.tw (S.D.); adrian.santoso@alumni.i3l.ac.id (A.W.S.);; 2Department of Emergency Medicine, Kaohsiung Medical University Hospital, Kaohsiung Medical University, Kaohsiung 80708, Taiwan; yohoco@gmail.com; 3Graduate Institute of Clinical Medicine, College of Medicine, Kaohsiung Medical University, Kaohsiung 80708, Taiwan; 4Ph.D. Program for Cancer Molecular Biology and Drug Discovery, College of Medical Science and Technology, Taipei Medical University and Academia Sinica, Taipei 11031, Taiwand621109004@tmu.edu.tw (H.D.K.T.); 5Department of Biotechnology, University of Health Sciences, Lahore 54600, Pakistan; waqarmazhar63@gmail.com; 6TMU Research Center of Cancer Translational Medicine, Taipei Medical University, Taipei 11031, Taiwan

**Keywords:** glioblastoma, genome, malignancy, G-protein-coupled receptor (GPCR), molecular biomarker, prognosis, signaling pathway

## Abstract

**Simple Summary:**

In this study, we aimed to address the critical need for a prognostic biomarker in the treatment of GBM. Various approaches and treatments have been examined in the recent literature; however, their effectiveness is limited due to the highly invasive, heterogeneous, and resistant nature of GBM tumors. TCGA, which is a database available online, was used to assess the role of guanine nucleotide-binding protein G(i) subunit alpha 3 *(GNAI3)*, with a focus on analyzing its impact across different WHO grades. The results revealed that *GNAI3* is associated with a poor prognosis and is involved in significantly important pathways, such as macrophage maturation and cytoskeleton arrangements. These findings suggest that *GNAI3* may serve as a valuable prognostic biomarker for the GBM microenvironment and could provide actionable information for the treatment of GBM.

**Abstract:**

Glioblastoma multiforme (GBM) is one of the most aggressive cancers with a low overall survival rate. The treatment of GBM is challenging due to the presence of the blood–brain barrier (BBB), which hinders drug delivery. Invasive procedures alone are not effective at completely removing such tumors. Hence, identifying the crucial pathways and biomarkers for the treatment of GBM is of prime importance. We conducted this study to identify the pathways associated with GBM. We used The Cancer Genome Atlas (TCGA) GBM genomic dataset to identify differentially expressed genes (DEGs). We investigated the prognostic values of the guanine nucleotide-binding protein G(i) alpha subunit (*GNAI*) family of genes in GBM using a Chinese Glioma Genome Atlas (CGGA) dataset. Within this dataset, we observed the association in the tumor microenvironment between the gene expression of GNAI subunit 3 (*GNAI3*) and a poor prognosis. MetaCore and gene ontology (GO) analyses were conducted to explore the role of *GNAI3* in co-expressed genes and associated signaling pathways using a transcript analysis. Notable pathways included “Cytoskeleton remodeling regulation of actin cytoskeleton organization by the kinase effectors of Rho GTPases” and “Immune response B cell antigen receptor (BCR) pathway”. A single-cell analysis was used to assess GNAI3 expression in GBM. The results demonstrated that *GNAI* family genes, specifically *GNAI3*, were significantly associated with carcinogenesis and malignancy in GBM patients. Our findings suggest that the *GNAI3* gene holds potential as a prognostic biomarker for GBM.

## 1. Introduction

Glioblastoma, commonly known as glioblastoma multiforme (GBM), is an exceedingly aggressive and lethal type of brain tumor that originates from glial cells, which are supportive cells in the brain [1]. GBM usually develops in the cerebral hemispheres, the main regions of the brain responsible for processes such as memory, speech, and movement. It can also expand to other parts of the central nervous system (CNS) and other organs in rare circumstances [2]. Research suggests that the progression of GBM is influenced by genetic mutations and abnormal changes in DNA, although the precise underlying cause remains under investigation [3,4]. GBM manifestations vary, depending on the tumor size and location. Common symptoms include persistent headaches, seizures, cognitive impairments, alterations in personality or behavior, limb weakness or numbness, challenges with speech or vision, and episodes of nausea or vomiting [5]. A comprehensive evaluation of a patient’s medical history, a neurological examination, the use of imaging techniques such as magnetic resonance imaging (MRI) or computed tomography (CT), and a biopsy procedure where a sample of the tumor tissue is extracted for further analysis are all aspects of the diagnostic process for GBM [6]. Treatment usually consists of a combination of surgical resection (tumor removal), radiation therapy, and chemotherapy [7]. The primary goal of surgery is to remove as much of the tumor as possible whilst minimizing damage to healthy brain regions [8]. Radiation therapy is used to target and eliminate or slow the growth of cancer cells [9]. Yet despite these rigorous treatment techniques, GBM remains difficult to completely cure due to its infiltrative nature and high recurrence rate [10]. Patients have poor prognosis, with a median survival period of 12 to 15 months after diagnosis [11]. Thus, treatment advances and ongoing research is hoped to provide the potential for better outcomes and quality of life for GBM patients.

There is a growing interest in investigating the functions of individual proteins within cellular signaling pathways and their effects on various diseases [12]. The *GNAI* family, with a particular focus on the guanine nucleotide-binding protein G(i) subunit alpha 3 (*GNAI3*), represents a noteworthy protein family of interest [13]. *GNAI* proteins are important components of the larger family of G proteins, performing crucial functions in transducing signals from receptors on the cell surface to the interior of cells and regulating a wide range of cellular processes [14]. The *GNAI* family comprises the following three major members: *GNAI1*, *GNAI2*, and *GNAI3*. These proteins decrease cellular signaling pathways by lowering the activity of adenyl cyclase, the enzyme responsible for producing the second messenger molecule cyclic adenosine monophosphate (cAMP) [15]. *GNAI* proteins have a variety of activities in several tissues and cell types, including neurotransmission, hormone signaling, cell growth, differentiation, and motility. The dysregulation of and mutations in *GNAI* proteins have been linked to a variety of diseases and disorders. Alterations in *GNAI3* are associated with specific types of pituitary tumors and difficulties with excesses of growth hormones [16]. *GNAI2* mutations have been identified in specific somatic overgrowth syndromes [17]. The dysregulation of *GNAI* proteins has also been identified in a variety of cancers, including breast and colorectal cancers; these proteins were reported to influence tumor growth and metastasis [18]. In short, the *GNAI* protein family, including *GNAI1*, *GNAI2*, and *GNAI3*, exhibits a versatile nature that underscores its importance in cellular signaling and regulation, contributing to a range of physiological and pathological functions.

*GNAI3* encodes the *GNAI3* protein and exhibits functional activity in multiple tissues, including the brain, the heart, and skeletal muscle [19]. Extensive research has provided an insight into the activities of *GNAI3* and its implications in cellular processes and signaling pathways. It plays a critical function in G-protein-coupled receptor (GPCR) signaling. It is activated when GPCRs are activated by ligands, detaching from the receptor and participating in downstream signaling cascades [20]. It is also engaged in vascular smooth muscle cell contractions; its activation inhibits calcium signaling pathways, resulting in blood vessel relaxation and dilation [21]. These findings illustrate how *GNAI3* affects blood vessel functions and overall vascular health. It modulates neurotransmitter release in the CNS and regulates synaptic transmission by suppressing adenyl cyclase activity and reducing cAMP levels in neurons [22]. This mechanism emphasizes the importance of *GNAI3* in fine-tuning neurotransmission and sustaining appropriate neuronal signaling. The dysregulation of *GNAI3* has been linked to malignancies. Its aberrant expression or genetic mutations have been detected in different cancer types, including breast cancer, colorectal cancer, and lung adenocarcinoma [23]. In these contexts, its dysregulation can impact tumor growth, metastasis, and response to therapy. Therefore, understanding the role of GNAI3 in carcinogenesis offers the potential for the development of targeted therapeutics and personalized medicine techniques.

The intricate relationship between GBM and the *GNAI3* protein family presents a novel study avenue. The aggressive nature of GBM and the multifaceted functions of *GNAI3* emphasize the requirement for the further exploration of their interactions and potential therapeutic interventions. Understanding the role of *GNAI3* in cellular processes and its implications for diseases, including GBM, may inspire innovative treatment strategies and ultimately improve patient outcomes. In this study, we aimed to establish a connection between *GNAI3*, its regulated pathways, and immune infiltration in GBM. Our goal was to identify a novel potential biomarker for immunotherapy. Our approach involved a comprehensive bioinformatics analysis, encompassing expression levels, pan-cancer assessments, overall survival (OS), the molecular function (MF), biological process (BP), the cellular process (CP), immune infiltration, MetaCore, and a single-cell analysis. This analysis revealed the involvement of *GNAI3* in critical BPs that can drive cancer progression and motility. We outlined a systematic workflow for a comprehensive survey of various genes encoding the *GNAI* family of proteins, with a focus on the potential candidates for GBM. This study presents our interpretation of the expression levels and elucidation of molecular mechanisms, along with function and enrichment analyses (Figure 1). 

## 2. Materials and Methods

### 2.1. Use of UALCAN for Pan-Cancer and Gene Expression Analyses

UALCAN (http://ualcan.path.uab.edu (accessed on 1 May 2023)) is an online data platform with clinical data for approximately 30 different types of cancer. It offers easy access to freely available cancer omics using CSS and JavaScript. The RSEM algorithm was employed for an RNA sequence analysis to determine the expression values of the genes in this database. A transcripts per million (TPM) measure was used to assess gene significance across various cancer types. These data were retrieved from TCGA. We focused on GNAI family genes in tumor samples (*n* = 156) and normal samples (*n* = 5) from GBM patients. The expressions of GNAI family genes were observed using the GBM dataset from TCGA, with a *p*-value of <0.05 [24].

### 2.2. Chinese Glioma Genome Atlas (CGGA) Dataset for Survival Analysis

The CGGA (http://www.cgga.org.cn/ (accessed on 1 May 2023)) database is an online open-access platform, with clinical data and sequencing of more than 2000 brain tumor samples. The CGGA dataset was accessed to determine the distribution of the messenger RNA (mRNA) expression levels of *GNAI* family genes for World Health Organization (WHO) grades I, II, and III as well as the expressions of *GNAI2* and *GNAI3* between isocitrate dehydrogenase (IDH) mutant and wild-type tumor samples. An OS analysis was performed for both primary and recurrent gliomas using a significant *p* value of <0.05. Survival analyses for *GNAI2* and *GNAI3* were performed using the mRNAseq_325 platform, which contains mRNA sequencing data from a total of 325 samples (pediatric low-grade gliomas = 144, recurrent low-grade gliomas = 38, progressive GBM = 85, recurrent GBM = 24, and secondary GBM = 30), using the Illumina high-throughput sequencing platform [25].

### 2.3. Protein–Protein Interactions (PPIs)

PPIs (https://string-db.org/ (accessed on 1 May 2023)) were assessed using a STRING analysis, an online web tool that contains data on more than 10 different types of organisms (including humans), with 19,303 protein linkage connections. We performed an analysis of *GNAI* family genes for *Homo sapiens*. This analysis provided an illustrative view of the protein interactions.

### 2.4. Drug Sensitivity

To understand the drug sensitivity of our specific genes, we used Genome Set for Cancer Analysis (GSCA) (http://bioinfo.life.hust.edu.cn/GSCA/ (accessed on 1 May 2023)), an integrated online tool that integrates TCGA data for 30 cancer types with pharmacogenomic and immunogenic cancer analyses. GSCA offers two types of drug datasets: (1) the Genomics of Drug Sensitivity in Cancers (GDSC), which provides a list of drugs and their correlations with sensitivities to specific gene expressions using 50% inhibitory concentration (IC_50_) values, and (2) the Cancer Therapeutics Response Portal (CTRP), which contains data on the small molecules that target specific genes or pathways [26].

### 2.5. Human Protein Atlas

We used the Human Protein Atlas (https://www.proteinatlas.org/ (accessed on 1 May 2023)) to investigate the protein expressions of GNAI2 and GNAI3 in GBM. The immunohistochemistry images included clinico-pathological information, such as the patient’s age, gender, and ID, as well as normal and tumor samples. We also observed the expression of GNAI3 in different GBM cancer cell lines [27].

### 2.6. Functional Enrichment GO (Gene Ontology) Analysis

The cBioPortal (https://www.cbioportal.org (accessed on 1 May 2023)) is an online database with various datasets that provide a statistical representation of gene correlations and co-expressions using mRNA expressions. This database was accessed to obtain the GBM dataset from TCGA (*n* = 592 samples). Co-expression gene data, such as BPs, MFs, and CFs, were used to conduct a GO (gene ontology) enrichment analysis using Spearman’s correlations. The enrichment analysis also encompassed the Kyoto Encyclopedia of Genes and Genomes (KEGG) pathways, which demonstrate relationships and abundances among pathways [28]. The fold enrichment of the x-axis was calculated using a −10 log (*p* value) in dot plots for the pathways [29]. The enrichment analysis was conducted using the R software clusterprofiler tool (R 4.3.1 version), which was designed for BPs, MFs, CFs, and the KEGG [30]. *We* also used the MetaCore analysis (https://portal.genego.com/ (accessed on 1 May 2023)) to analyze the cell signaling pathway [31].

### 2.7. TIMER 2.0 for Immune Infiltration

Using TIMER 2.0 (http://timer.comp-genomics.org/ (accessed on 1 May 2023)), we explored the infiltration of immune cells in 31 cancer types from TCGA datasets. This tool uses Spearman’s correlation coefficients and statistical significance. We studied the correlations between the highly expressed *GNAI3* gene and the infiltration of inflammatory cells in GBM for immune infiltration in default immune cells, such as clusters of differentiation 4-positive (CD4+) T cells, CD8+ T cells, B cells, macrophages, dendritic cells (DCs), and neutrophils.

### 2.8. Single-Cell Analysis

We conducted a single-cell analysis using CancerSEA (http://biocc.hrbmu.edu.cn/CancerSEA/home.jsp (accessed on 1 May 2023)) [32]. The CancerSEA database contains 430 cells from individual tumors and 102 cells from gliomasphere cell lines generated using SMART-seq. It retrieves information from sources such as the Gene Expression Omnibus (GEO) and TCGA. Our analysis encompassed small conditional (SC) RNA expression across 25 different cancers, taking heterogenicity into consideration. We also explored the PCG/long non-coding (LNC) RNA repositories. The database provides single-cell analytical data for different functional states. Meanwhile, we conducted an analysis using the Single-Cell Portal (https://singlecell.broadinstitute.org/single_cell (accessed on 1 May 2023)) database for GBM in adults and pediatrics. This revealed the expressions of *GNAI3* in different single-cell clusters. The Single-Cell Portal is an online repository that gathers data from TCGA and GEO. The dataset used for the single-cell analysis was “single-cell RNA-seq analysis of adult and pediatric IDH-wild-type glioblastomas”, which comprises 24,131 cells from 28 tumor samples.

### 2.9. Statistical Analysis

The expression levels of genes were evaluated using Student’s *t*-test, Spearman’s correlation analysis, and Pearson’s correlation test. Survival curves from the CGGA were compared using a log-rank test, and statistical significance was determined when *p* < 0.05. The results of the log-rank analysis are presented as a *p* value and a hazard ratio (HR). 

## 3. Results

### 3.1. Expression of GNAI3 in a Pan-Cancer Analysis of DEGs Using TCGA Data

TCGA datasets were used to analyze the differential expressions of the GNAI family genes across all human cancers. The pan-cancer analysis revealed the expressions of *GNAI*1–3 in 20 different cancer types from the UALCAN database. The results demonstrated that *GNAI1* was not significantly or differentially expressed in normal or tumor tissues, whereas *GNAI2* expression was higher in GBM and sarcomas. The pan-cancer mRNA expression of *GNAI3* was higher in breast-invasive carcinomas, cervical squamous cell carcinomas, esophageal carcinomas, head and neck squamous cell carcinomas, glioblastomas, sarcomas, and skin cutaneous carcinomas. UALCAN was used to obtain box plots to identify the expressions of GNAI family genes and their significance using the GBM dataset from TCGA. Among all family members, *GNAI2* and *GNAI3* demonstrated differential expression levels between the GBM tumor and normal tissue samples, whereas *GNAI1* was not significantly or differently expressed. Collectively, this demonstrated that *GNAI2* and *GNAI3* had higher mRNA expression in GBM than in normal tissues (Figure 2). Therefore, we focused on *GNAI2* and *GNAI3* because they demonstrated a higher expression in GBM than in the other cancer types.

### 3.2. Expressions of GNAI2 and GNAI3 in IDH WTs Based on WHO Grade II, III, and IV Gliomas and Survival Analysis

We performed an analysis using the mRNA_325 dataset from the CGGA database to determine the expressions of *GNAI* family genes in WHO glioma grades II, III, and IV. The results revealed that there was a significant gradual increase in the expression levels from grades II to IV GBM for *GNAI2* and *GNAI3* (*p* = 7.7~15 and *p* = 1.1~12, respectively). We conducted an analysis to assess the impact of the IDH mutation status on the expression levels of *GNAI2* and *GNAI3* using a *t*-test with statistical significance set at *p* < 0.05. The results revealed that the mRNA expression was slightly higher among the WTs than the mutant IDH samples. *GNAI2* demonstrated a gradual increase in expression levels among the IDH WTs and was highly significant for WHO grade IV types. *GNAI3* also demonstrated a gradual increase in expression levels in grade III and IV IDH WTs, whereas the expression of mutant IDH in the WHO grade II samples revealed that the effect of IDH was significant (Figure 3) [33]. We examined the expression levels of *GNAI* family members and the OS rate using the CGGA database and mRNA_325 dataset for primary and recurrent gliomas, respectively. The results indicated that a higher GNAI2 expression was associated with a poorer overall prognosis in both primary and recurrent gliomas; this association was statistically significant (*p* < 0.0001) in primary gliomas. *GNAI3* also had a poor prognosis and disease-free survival (DFS); this was statistically significant in both primary (*p* < 0.0001) and recurrent gliomas (*p* = 0.0017). The analysis of *GNAI* family expression profiles clearly demonstrated that elevated gene expression levels were linked to shorter disease-free survival (DFS) in patients, as illustrated in Figure 4.

### 3.3. PPI Analysis

We investigated PPIs using the STRING-DB online web tool. The network consisted of 41 nodes and 572 edges. It revealed that *GNAI* family genes interacted with each other, and the other members of the guanine nucleotide-binding (GNB) family were involved in cell cycle responses and signaling pathways (Figure 5).

### 3.4. Drug Analysis and the Role of Drugs in GBM Treatment

Using a GDSC dataset, GNAI2 demonstrated sensitivity to and correlated with afatinib, whereas GNAI3 demonstrated a greater sensitivity to dabrafenib. The CTRP data revealed that lapatinib, austocystin D, and afatinib were effective for GNAI2, whereas linsitinib was effective for GNAI3. All of these drugs are tyrosine kinase inhibitors (TKIs), except for austocystin D. TKIs target the epidermal growth factor receptor (EGFR) and Erb-B2 Receptor Tyrosine Kinase 2 (ERRB2) pathways, which are known to play crucial roles in tumor progression and metastasis. Austocystin D functions as an inducer of DNA damage and targets nuclear receptor subfamily 1 (Figure 6). The GDSC and CTRP results revealed that TKIs stunted cell progression and metastasis.

### 3.5. Human Protein Atlas (HPA) Analysis Results

After performing the survival analysis, the protein expressions of *GNAI2* and *GNAI3* were examined for GBM using immunohistochemical (IHC) scores from HPA. The IHC images included clinico-pathological information, such as the patient’s age, gender, and ID, as well as normal and tumor samples. In the analysis, the *GNAI2* protein was not significantly or differentially expressed between the normal and tumor tissues. The expression of the *GNAI3* protein was higher in the tumor tissues. The protein expression levels were quantified using a score that was automatically determined based on a combination of the staining fraction and intensity. The scoring system included categories such as negative (not detected), weak (<25% (low)), moderate (<25% (medium)), and strong (<25% (medium) to <25% (high)), with corresponding definitions for combined fractions and intensities.

The analysis of the HPA results revealed a strong association between GNAI3 and GBM in the staining samples. The expression of GNAI2 was less distinct, suggesting that it may have a limited role as a prognostic biomarker for GBM. 

The expression of *GNAI3* in brain cancer cell lines was observed to be notably higher and more significant in the DOAY, U87MG ATCC, and SF172 cell lines than in the other cell lines (*n* = 65), as shown in Figure 7. 

Based on the IHC staining and differential expression between normal and cancer tissues, we focused on *GNAI3* for further analysis. *GNAI3* demonstrated statistical significance in primary and recurrent gliomas; consequently, we conducted an in-depth analysis to assess its role as a potential prognostic biomarker for GBM because it demonstrated a high mRNA expression that was highly significant. It also exhibited an increased expression as the tumor grew; a higher expression of this gene led to poor survival outcomes due to the poor prognostic value.

### 3.6. Pathway Interactions Using GO Tools for GNAI3 

To understand the role of GNAI3, TCGA GBM data were retrieved to determine the GO results for MFs, CFs, BPs, and the KEGG. A gene ontology (GO) analysis was conducted using the R software 4.3.1 clusterprofiler package for a pathway analysis. Spearman’s correlation (>0.45) was used as a threshold; 127 genes were input for the GO analyses using a co-expression dataset obtained from cBioPortal. GNAI3 was associated with catalytic activity, acting on RNA and ribonucleoprotein complex binding in the MFs. *GNAI3* demonstrated an involvement in nuclear speckles and spliceosomal complexes in the CCs. The BP pathways included ribonucleoprotein complex biogenesis, ribosome biogenesis, non-coding (nc) RNA processing, and RNA splicing. The associated KEGG pathways included spliceosomes, nucleo-cytoplasmic transport, and ribosome biogenesis in eukaryotes (Figure 8). These results revealed that *GNAI3* had involvement in cellular responses and cell signaling processes.

### 3.7. TIMER Analysis for Immune Infiltration

We used the TIMER database to determine immune infiltration associated with *GNAI3* in GBM. The tumor microenvironment (TME) is a changing and emergent entity; it varies among cancer types. Research on the TME is ongoing, but existing research has demonstrated the importance of the TME in cancer growth and progression [34]. To understand the effects of immune cells, we performed a TIMER analysis of *GNAI3*. It demonstrated a positive correlation with myeloid DCs (Rho = 0.283; *p* = 8.00 × 10^−4^). It was negatively correlated with B cells (Rho = 0.055; *p* = 5.23 × 10^−1^), CD4+ T cells (Rho = −0.01; *p* = 9.12 × 10^−1^), macrophages (Rho = 0.091; *p* = 2.93 × 10^−1^), neutrophils (Rho = 0.145; *p* = 9.14 × 10^−2^), and CD8+ T cells (Rho = 0.029; *p* = 7.33 × 10^−1^) (Figure 9). The role of B cells varies in each cancer type; growth is enhanced by B cells, which can induce an immunosuppressive response by the activation of inhibiting FC receptors on myeloid cells [35].

The MetaCore platform was used to identify the BP pathways stimulated by *GNAI3* using the co-expressed gene dataset from TCGA. It revealed that *GNAI3* was involved in various significant pathways that regulate BPs, such as “cytoskeleton remodeling regulation of actin cytoskeleton organization by kinase effectors of Rho GTPases” (Figure 10), “DNA damage double-strand break repair via homologous recombination”, “immune response B cell antigen receptor (BCR) pathway”, “G protein signaling Ras-family GTPases in kinase cascades”, “development and regulation of telomere length and cellular immortalization”, “cytoskeleton remodeling regulation of actin cytoskeleton nucleation and polymerization by Rho GTPases”, “DNA damage ATM/ATR regulation of G2/M checkpoint: cytoplasmic signaling”, “development regulation of cytoskeleton proteins in oligodendrocyte differentiation and myelination”, “cell cycle DNA replication initiation”, and “immune response ETV3 effect on CSF1-promoted macrophage differentiation”. These findings provide better understanding of cancer development in GBM (Appendix A).

### 3.8. GNAI3 Expression Using Single-Cell Analysis

A single-cell analysis was performed to identify the unique gene expression patterns or mutations associated with specific cell types in cancer [36]. These findings could lead to the discovery of biomarkers that could be used for the early diagnosis or prognosis of GBM. The single-cell analysis was performed using two different online portals, yieldng similar results. The CancerSEA database revealed a highly correlated and significant role of GNAI3 in DNA repair functions (Figure 11). Meanwhile, the Single-Cell Portal revealed the chromosomal location of GNAI3 and co-expressed genes. It also indicated that GNAI3 expression was relatively high in macrophages compared to other cells, such as oligodendrocytes, T cells, and malignant cells (Figure 12).

## 4. Discussion

GBM is the most common and lethal malignant primary brain tumor in adults, accounting for 54% of all gliomas and 16% of all primary brain tumors [37]. Despite intensive treatments using safe surgical excision, radiation, and chemotherapy methods, the median survival is only approximately 14.6 months, and the five-year survival rate is <10% [38]. To date, the Stupp regimen remains the standard treatment for GBM. This includes surgery where possible, followed by concurrent treatments of radiation and chemotherapy with temozolomide (TMZ); afterward, TMZ alone is administered. Other treatment strategies are also being explored for GBM treatment. Immunotherapies such as checkpoint inhibitors, chimeric antigen receptor (CAR) T-cell therapy, and vaccines are being studied. Considerably, the brain’s immune system is unique and therefore can be a challenging location for immunotherapy. Glioblastoma stem cells (GSCs) have recently become the focus of GBM research [39]. These cells are known to be resistant to standard therapy and may be at the root of tumor recurrence. Drugs intended to suppress growth signals (such as EGFR inhibitors) or angiogenesis inhibitors to cut off the blood supply to tumors are other examples of ongoing research areas [40]. Recent research has revealed that TKIs play significant roles in tumor progression management. They have been revealed to trigger autophagy in neuroblastoma (NB) cells, resulting in a significant increase in cell death [41]. In this study, we present an overview of how *GNAI* family proteins, particularly *GNAI3*, influence tumor growth and progression, as well as immune responses, with the potential to function as a cancer biomarker. The *GNAI3* protein is a member of the G protein family, which is important in cell signal transduction. Alterations in these signaling pathways can disrupt cellular processes, such as cell proliferation, differentiation, migration, and apoptosis, leading to tumorigenesis. The *GNAI3* protein may be involved in various cancer-related processes.

*GNAI3* may alter cell migration and invasion, two important mechanisms involved in tumor metastasis, in the context of tumor progression. GPCR signaling in *GNAI3* plays an important role in controlling immune cells in immune responses. Any disruption in this pathway could impair the immune system’s capacity to effectively react to malignancies [42]. In terms of growth, G proteins are important in signal transduction from growth factor receptors to downstream effectors. Abnormalities in these signals can lead to uncontrolled cell growth, a hallmark of cancer. The relevance of *GNAI3* as a cancer biomarker is mostly determined by whether its expression or level of activity is consistently associated with certain features of cancer, such as its existence and stage, as well as its response to treatment.

We investigated *GNAI3* expression in various cancer types and its correlation with OS, a poor prognosis, and immune infiltration in GBM using TCGA. These behaviors were investigated through bioinformatics research using *GNAI3* as a prognostic biomarker for GBM. It was observed that *GNAI3* expression gradually increased in WHO grade II–IV gliomas and that IDH mutations were associated, indicating that *GNAI3* expression was not affected. A GO pathway study revealed that *GNAI3* plays a critical role in cell signaling, ribosomal synthesis, and mRNA splicing. The involvement of *GNAI3* in CPs and tumor progression has promoted additional bioinformatics research into *GNAI3*. 

In terms of the involvement of *GNAI3* in the aforementioned pathways, Rho GTPases are critical in cytoskeletal remodeling and actin regulation. Actin (along with myosin) is a protein that produces contractile filaments of muscle cells and is involved in several cellular movements, including cell motility, cell structure and integrity, and intracellular trafficking. Cellular transformation, including the genesis and growth of cancer cells, is typically accompanied by changes in the cytoskeleton and organization of the actin structure. *GNAI3* may play a role in the onset or progression of GBM by altering cellular structures and behaviors [43]. Significantly elevated or reduced expression levels in GBM may indicate a relationship between the gene and cancer; this could be used to detect or monitor the disease. The WHO score analysis includes consideration of how *GNAI3* expression is implicated in pathway changes, with GBM progression from lower (less severe) to higher (more severe) WHO scores. This may reveal how *GNAI3*, as a biomarker, necessitates the consideration of its expression levels in GBM compared with healthy tissues.

The positive correlation between *GNAI3* and DCs in the setting of infiltrating the immune cells of GBM may imply that enhanced *GNAI3* signaling promotes DC recruitment or activity. DCs are antigen-presenting cells (APCs) that play a key role in initiating the adaptive immune response. They may boost antitumor responses by delivering tumor antigens and activating T cells. Negative correlations of *GNAI3* with B cells, CD4+ T cells, CD8+ T cells, macrophages, and neutrophils imply that *GNAI3* signaling may impede the recruitment or function of those cells [44,45]. *GNAI3* is also involved in signal transduction. It is implicated in several intracellular signaling pathways; the B-cell antigen receptor (BCR) pathway is critical for the activation and function of B cells. When an antigen binds to the BCR, it initiates a series of intracellular signaling events that result in the activation, proliferation, differentiation, and antibody production of B cells. Notably, while the BCR and GPCR pathways are distinct, but they may overlap. Several studies have revealed that G proteins play a role in regulating BCR signaling, altering B-cell activation and function. Little is known about the involvement of *GNAI3* with B-cell function in the context of GBM. B cells are part of a complex immune response to GBM, but their specific roles and interactions in the GBM TME remain unknown. Additional research is required to determine how *GNAI3* affects B-cell functions and immunological responses in GBM [46]. APCs contribute to IL-1, IL-6, IL-8, and IL-10 signaling along with macrophages and DCs in innate immunity and tumor cells in senescence. They detect tumor antigens by detecting MHC class I deficiencies in tumor cells or via the uptake of antigens by dying tumor cells, which activates CD4+ T cells via the T-cell receptor (TCR), CD28, and B-cell responses. Specifically, this directly or indirectly activates type 2 helper T (Th2) cells. Th2 cells inhibit malignancies by releasing cytokines, such as interferon (IFN)-γ. Th1 cells activate B-cell immunity and natural killer cells. IFN-γ also regulates M1 macrophages, which, in turn, kill cancer cells [47,48].

G proteins, including *GNAI3*, can regulate cell proliferation and survival; both are important in carcinogenesis and tumor progression. They also play roles in cell migration and invasion, which are critical for metastasis [49]. GPCRs, particularly *GNAI3*, are known to play roles in regulating immune cells in the setting of immunological responses. Alterations in this pathway may influence immune responses to GBM. Our study’s limitation is that we did not conduct a thorough verification using in vitro and in vivo experimental assays, but we believe that the findings of the bioinformatics analysis serve as a platform for future research. *GNAI3* has the potential to be a biomarker for immune infiltration in the development of GBM malignancies. Data from the single-cell analysis revealed a greater expression in macrophages, indicating that they play an important role in the malignancy of GBM [26]. A high expression in macrophages could be attributed to the fact that they are involved not only in the immune response but also in chemotaxis. Macrophages are involved in cellular responses and promote angiogenesis, as well as tumor cell migration and invasion [50]. We revealed that *GNAI3* could be used as a biomarker or a particular target in glioma immunotherapy. Although this is an early study, it is clear that there are significant limitations to how *GNAI3* may be altered and regulated with medicines. Further experiments should be performed to investigate the effects of *GNAI3* on immune cells within the TME in gliomas to develop personalized approaches such as vaccinations and to explore multiple clinical trials investigating immunotherapy combination studies.

## 5. Conclusions

Our results demonstrate that the overexpression of the *GNAI3* gene could be an indicator of a poor prognosis and a potential biomarker for GBM. A bioinformatics analysis, as well as MetaCore and single-cell analyses, revealed that it is involved in the cellular responses that trigger tumor progression and invasion. This biomarker could be the subject of future in-depth experimental validations. 

## Figures and Tables

**Figure 1 cancers-15-05112-f001:**
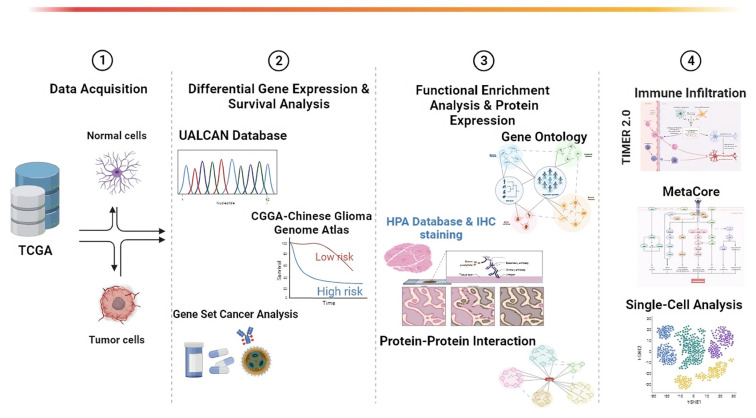
The workflow and experimental framework of the current study.

**Figure 2 cancers-15-05112-f002:**
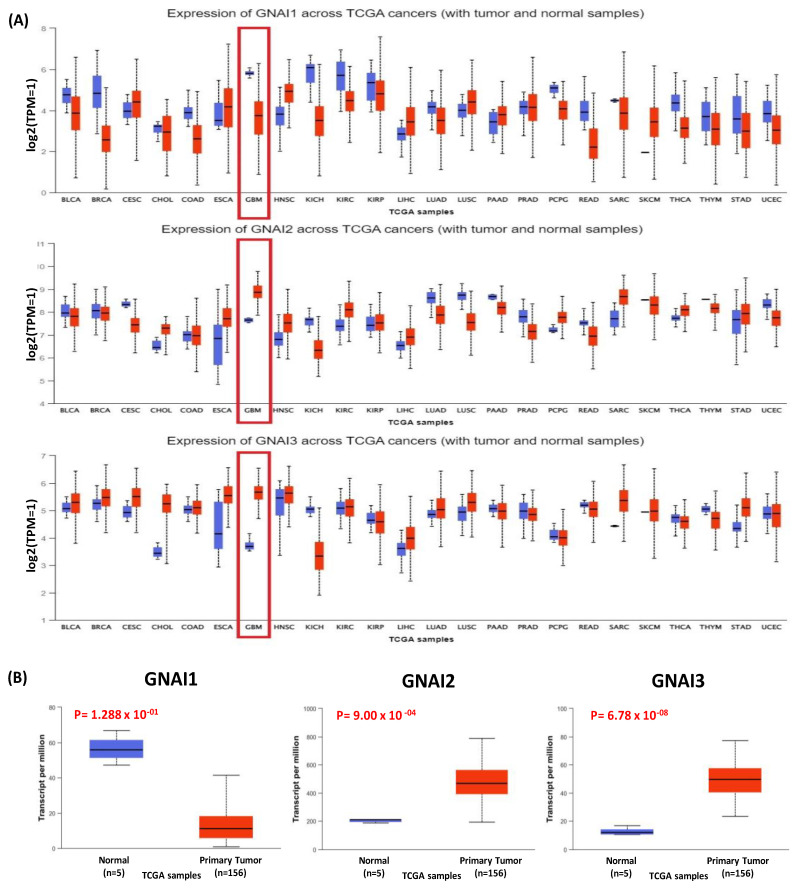
Differential expression analyses of *GNAI* family genes among cancers. (**A**) Expressions of *GNAI*1–3 across TCGA pan-cancers, revealing expression between primary tumors and normal tissues. (**B**) Box plot using TCGA GBM dataset from UALCAN, demonstrating *GNAI1* non-cancerous and both *GNAI2* and *GNAI3* cancerous samples; the significance was *p* < 0.05.

**Figure 3 cancers-15-05112-f003:**
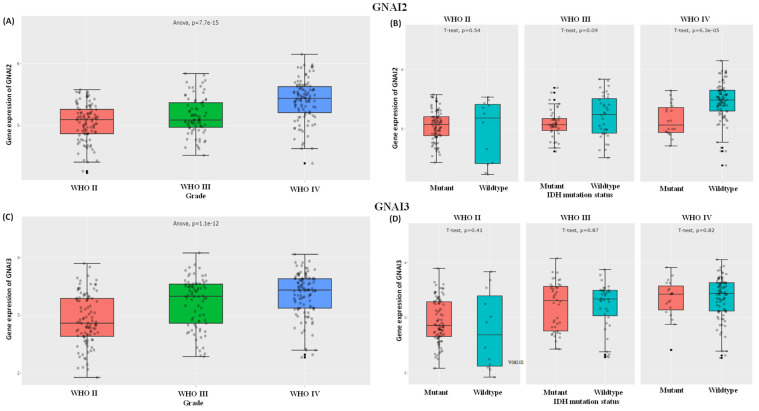
GNAI2/3 expression levels in GBM WHO grades. (**A**,**C**) The CGGA dataset was used for expressions of GNAI2 and GNAI3 in WHO grades II, III, and IV. (**B**,**D**) Expressions of GNAI2 and GNAI3 between IDH mutants and wild types in WHO glioma grades II, III, and IV.

**Figure 4 cancers-15-05112-f004:**
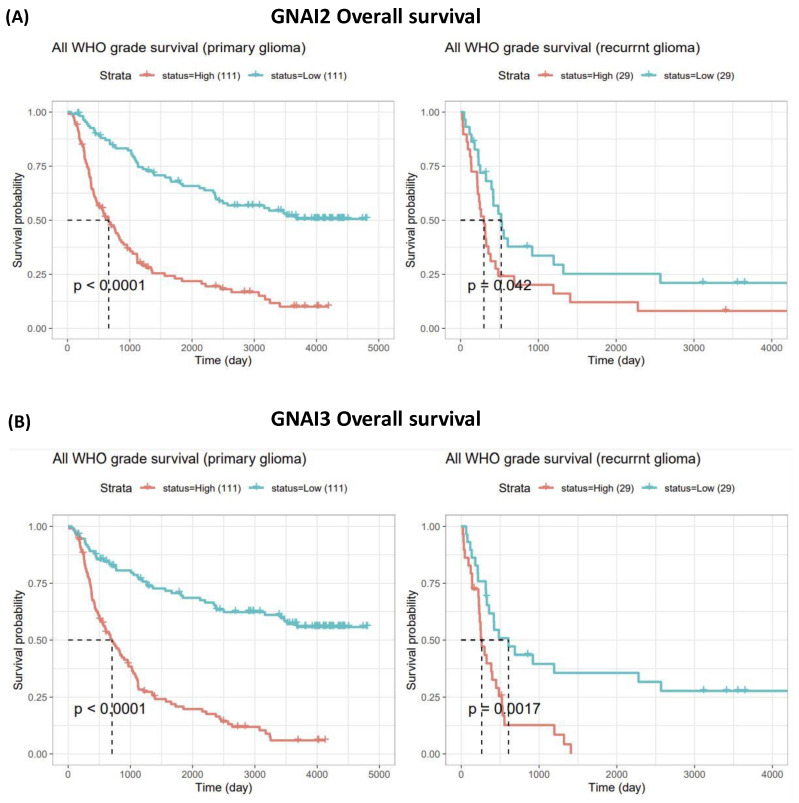
Survival prediction using the two-gene prognostic signature in primary and recurrent gliomas. (**A**) GNAI2 and (**B**) GNAI3 overall survival analysis revealed poor prognoses in both primary and recurrent gliomas, with significant *p* values < 0.05.

**Figure 5 cancers-15-05112-f005:**
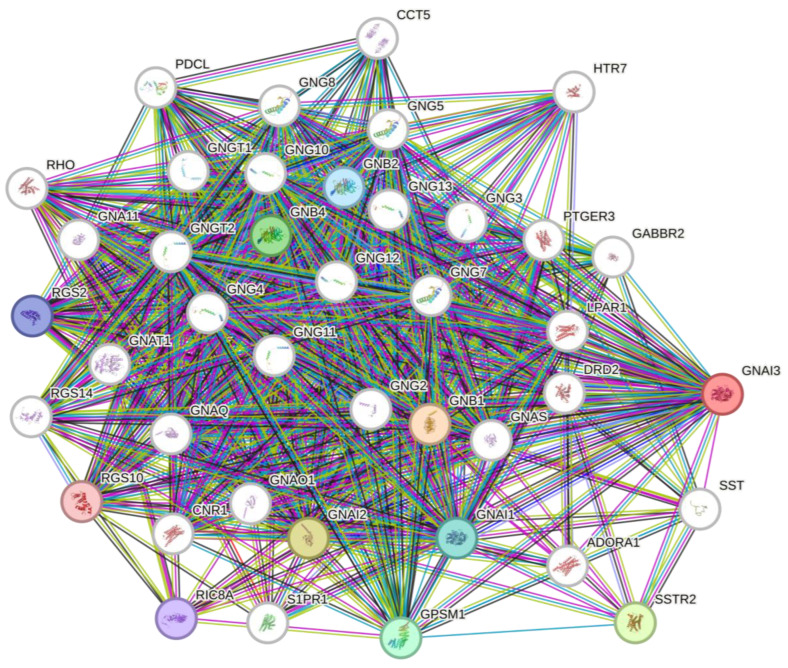
Protein–protein interactions of the *GNAI* family with the *GNB* protein family in cellular responses.

**Figure 6 cancers-15-05112-f006:**
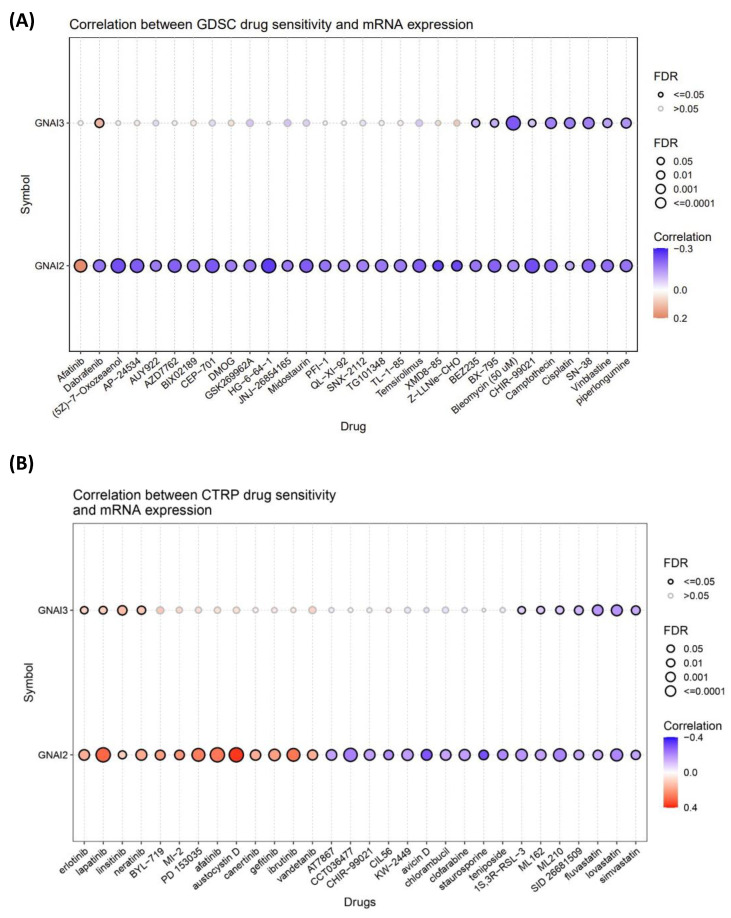
Drug sensitivity of GNAI2/3 oncogenes from GSCA. (**A**) Correlation between Genomics of Drug Sensitivity in Cancer (GDSC) data for FDA-approved drugs and GNAI2 and GNAI3. (**B**) CTRP drug data and sensitivity for *GNAI2* and *GNAI3*. The positive Spearman’s correlation coefficient (orange bubbles) indicated that an increased gene expression level was related to resistance to the drug compared with a negative correlation (shown in blue), which indicated the sensitivity of the drug.

**Figure 7 cancers-15-05112-f007:**
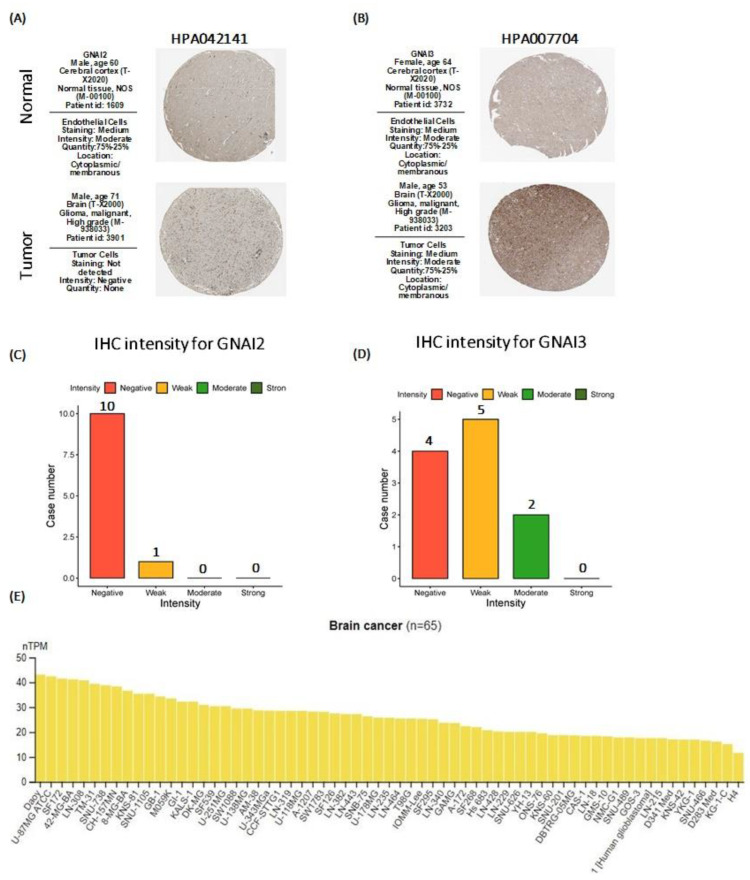
Representation of protein expression of *GNAI2/3* in GBM. (**A**,**B**) IHC staining images revealing expressions of the GNAI2 and GNAI3 proteins in normal and GBM tissues. (**C**,**D**) Bar graphs demonstrating the intensity of IHC staining for GNAI2 and GNAI3. (**E**) Expression of GNAI3 protein in brain cancer cell lines.

**Figure 8 cancers-15-05112-f008:**
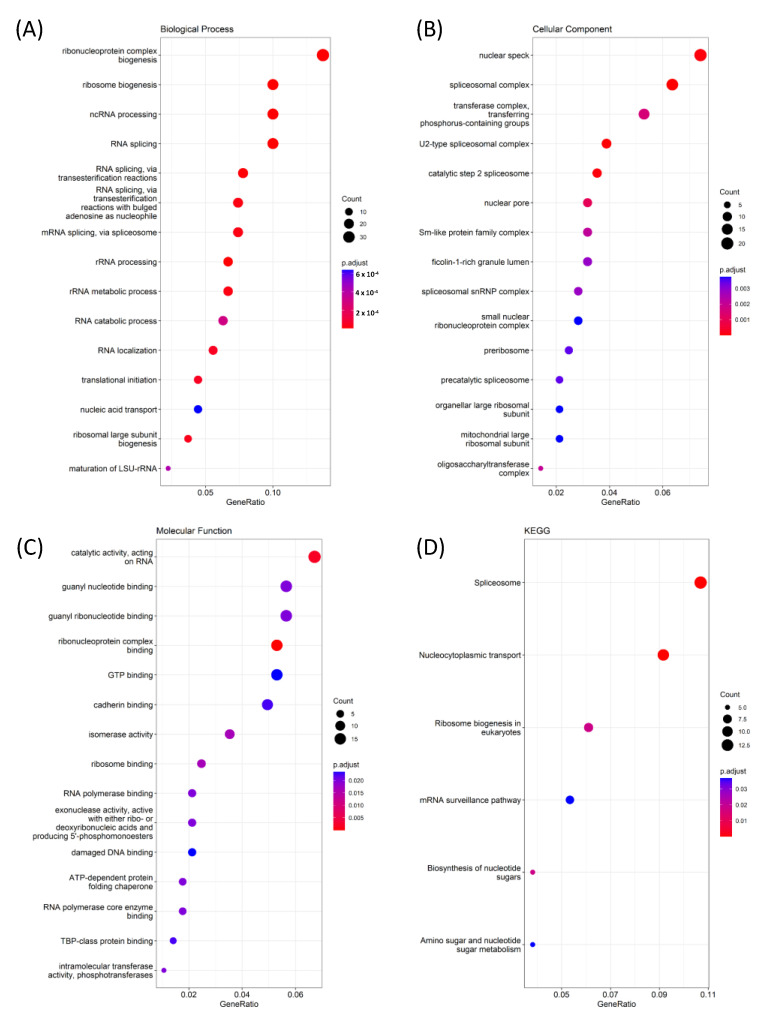
Gene ontology (GO) enrichment results using TCGA GBM differential gene expression analysis: genomic alterations of *GNAI3* in GBM from TCGA cohorts. (**A**–**C**) GO keywords with significant *p* values for genes: (**A**) BPs, (**B**) CCs, and (**C**) MFs for the *GNAI3* gene and upregulated genes. (**D**) Result of an enrichment analysis, including KEGG pathways with Spearman’s correlations > 0.45.

**Figure 9 cancers-15-05112-f009:**
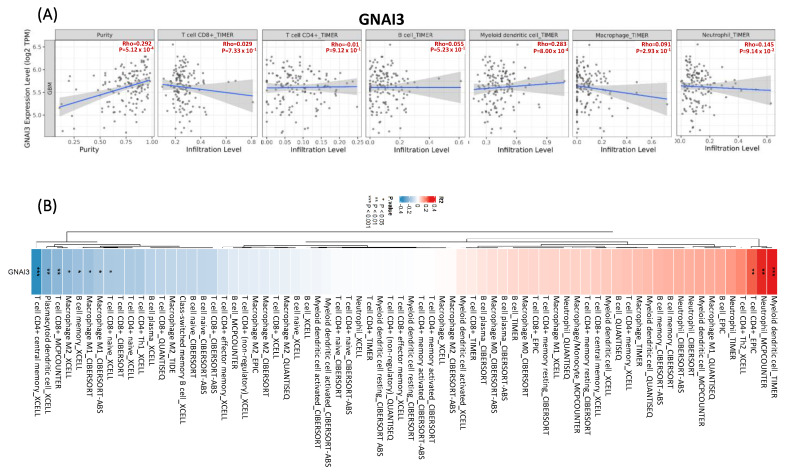
Relationship between *GNAI3* expression and immune infiltration of cells in GBM. The horizontal axis represents *GNAI3* expression (values represented as log2 RNA sequencing by expectation maximization (RSEM)); the vertical axis represents tumor-infiltrating immune cell markers. (**A**) *GNAI3* revealed a positive correlation with dendritic cells (DCs) and negative correlations with B-cell and cytotoxic T-cell immune responses. (**B**) Heat map of *GNAI3* demonstrating correlations with immune cells. 3.8. Expression of GNAI3 with Cytoskeleton Remodeling Regulation of Actin Cytoskeleton Organization by the Kinase Effectors of Rho GTPases

**Figure 10 cancers-15-05112-f010:**
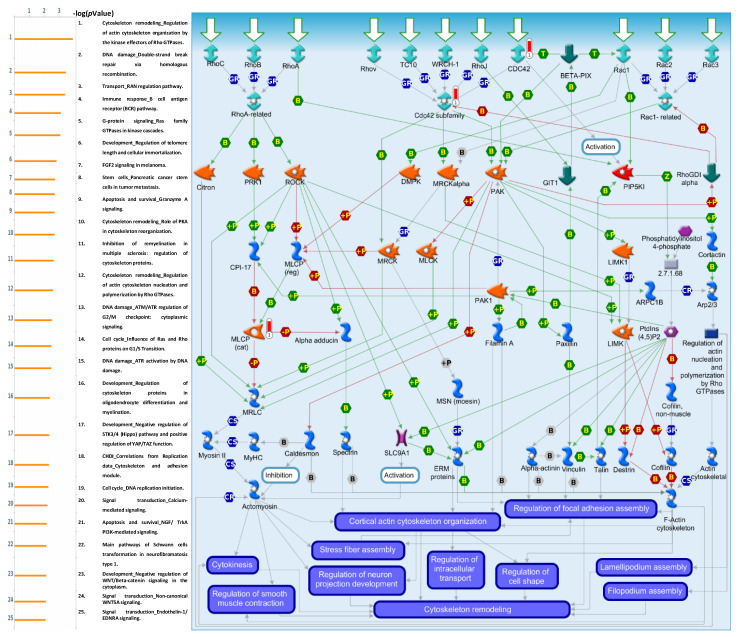
Expression of *GNAI3* signaling pathways in GBM using MetaCore. When analyzing the co-expression of genes of the *GNAI3* dataset from TCGA in MetaCore, we observed that *GNAI3*’s role in “cytoskeleton remodeling regulation of actin cytoskeleton organization by the kinase effectors of Rho GTPases” was associated with GBM progression (*p* < 0.05).

**Figure 11 cancers-15-05112-f011:**
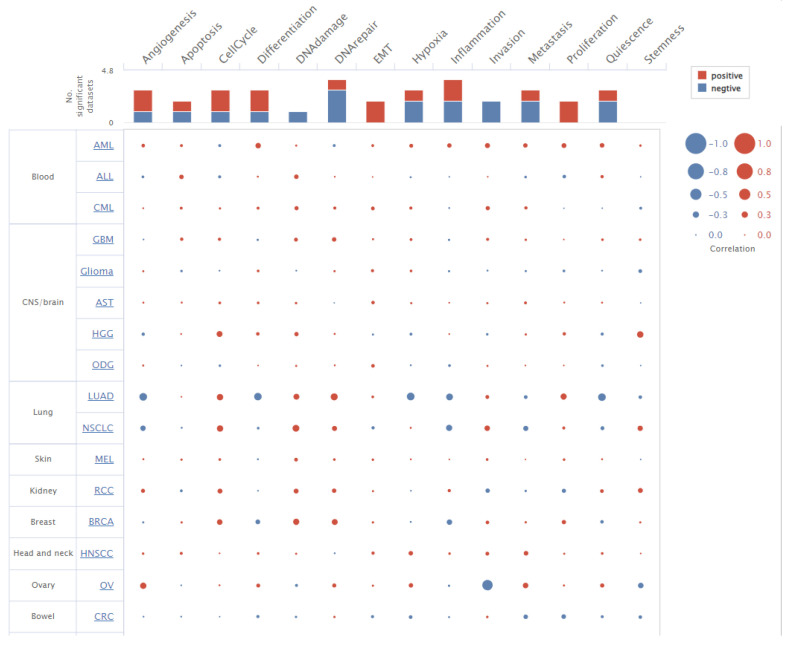
Single-cell analysis using the CancerSEA dataset demonstrated the role of *GNAI3* in different cell functions.

**Figure 12 cancers-15-05112-f012:**
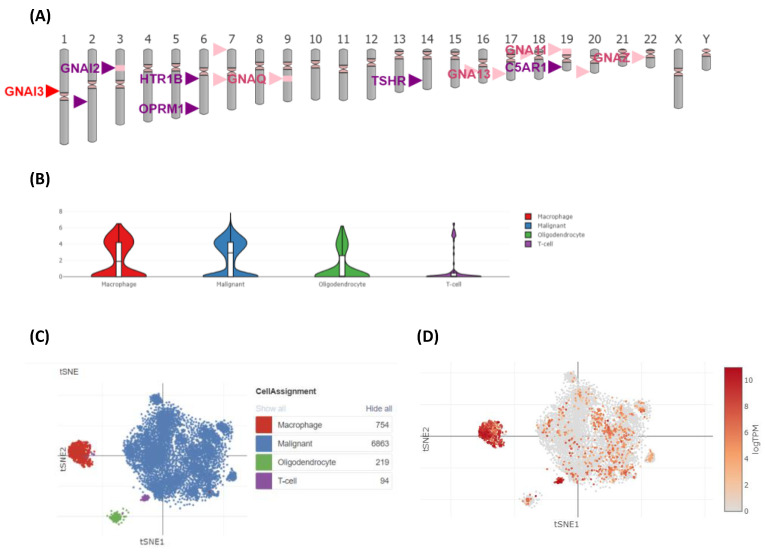
Single-cell analysis using the Single-Cell Portal. (**A**) Location of *GNAI3* and other correlated genes. (**B**) Violin plot demonstrating the expression of genes in different cell types. (**C**,**D**) tSNE plot demonstrating the expression of *GNAI3* in malignant cells, macrophages, oligodendrocytes, and T cells.

## Data Availability

All data supporting this study are available from the corresponding author upon reasonable request.

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
