# Peer review of "Comparative Analysis of the GNAI Family Genes in Glioblastoma through Transcriptomics and Single-Cell Technologies"

_cancers, 2023, doi:10.3390/cancers15205112_

Round 1

Reviewer 1 Report

Raza et al provide a comparative analysis of G protein alpha inhibiting subunit 3 in pathways associated with carcinogenesis and malignancy in patients with glioblastoma.  The authors synthesized a rationale for its participation in the evolution of glioblastoma including its association with immune modulation, the overall tumor microenvironment and its role in tumorigenesis and cancer proliferation.

Abstract including simple summary: Accurately described the hypothesis and methods involved

Introduction:  Provides an adequate background and the hypothesis provided on the role of the GNAI family, in particular GNAI3 in the formation and proliferation of glioblastoma.  Excellent background is provided as to the overall role of this gene in cancers in general, its role in the central nervous system and its potential role in tumor proliferation.

Materials and methods:  Adequately described the methods used in obtaining the information needed to address the hypothesis as detailed in introduction.

Results:  Adequately explains the results obtained.

Discussion:  To the best of my reading, the discussion is a well thought out synthesis and explanation of the findings as detailed in results, and provides an explanation of the findings and potential to translation for future study.  

Conclusions:  Adequately summarizes the basic finding that GNAI family gene overexpression could be considered an indicator of poor prognosis and potential biomarker for GBM, requiring additional studies.  Since this the study focused mostly on GNAI3, my suggestion would just be to address GNAI3 in the conclusion.  

References:  Adequate, reflects subject matter.

Figures and tables: No tables or figures redundant with text, all our complementary.  

Accept as is, editing for spelling and grammar is needed.

No issues, mostly misspellings 

Author Response

Dear Reviewer 1,

Comments and suggestions for authors:

Raza et al provide a comparative analysis of G protein alpha inhibiting subunit 3 in pathways associated with carcinogenesis and malignancy in patients with glioblastoma.  The authors synthesized a rationale for its participation in the evolution of glioblastoma including its association with immune modulation, the overall tumor microenvironment and its role in tumorigenesis and cancer proliferation.

Abstract including simple summary: Accurately described the hypothesis and methods involved

Introduction:  Provides an adequate background and the hypothesis provided on the role of the GNAI family, in particular GNAI3 in the formation and proliferation of glioblastoma.  Excellent background is provided as to the overall role of this gene in cancers in general, its role in the central nervous system and its potential role in tumor proliferation.

Materials and methods:  Adequately described the methods used in obtaining the information needed to address the hypothesis as detailed in introduction.

Results:  Adequately explains the results obtained.

Discussion:  To the best of my reading, the discussion is a well thought out synthesis and explanation of the findings as detailed in results, and provides an explanation of the findings and potential to translation for future study.  

Conclusions:  Adequately summarizes the basic finding that GNAI family gene overexpression could be considered an indicator of poor prognosis and potential biomarker for GBM, requiring additional studies.  Since this the study focused mostly on GNAI3, my suggestion would just be to address GNAI3 in the conclusion.  

References:  Adequate, reflects subject matter.

Figures and tables: No tables or figures redundant with text, all our complementary.  

Accept as is, editing for spelling and grammar is needed.:

Reply: We requested help from the Language Editing Services at MDPI and have made the necessary adjustments to ensure that the manuscript adheres to this writing style as it is the standard practice for scientific manuscripts. Your input helps to maintain the appropriate level of formality and objectivity in our work.

Sincerely,

Chih-Yang Wang, Ph.D (chihyang@tmu.edu.tw)

Graduate Institute of Cancer Biology and Drug Discovery,

College of Medical Science and Technology,

Taipei Medical University, Taipei, Taiwan.

Corresponding Authors.

Reviewer 2 Report

Minor remarks

1.     Avoid the use of abbreviations in the manuscript. All used abbreviations should be defined in the manuscript. Also, use the defined abbreviation throughout the whole manuscript.

2.     The manuscript should be written in the third-person singular. The first-person plural is not acceptable for the scientific manuscript.

3.     Many technical errors affect the quality of the manuscript.

4.     Each paragraph should be prepared in the same way. For instance, use the same line space, etc. Delete a blank line between two paragraphs.

5.     Avoid lumping the references. Each reference should be discussed separately.

6.     The figure order should be typed again.

7.     In some figures, the figure labels should be re-scaled to be readable.

8.      References should be prepared according to the Instructions for Authors.

9.      All minor remarks are depicted in the manuscript.

Major remarks

1.     I recommend the reduction of the references list. A literature review of older studies can be omitted from the manuscript. Avoid lumping the references. Each reference should be discussed separately.

2.     Also, the grammatical errors should be corrected in the manuscript.

3. The conclusion section is very poor. Please, retype this section and insert all the main conclusions of the research.

English language should be improved. There are some grammatical errors.

Author Response

       We would like to thank you for revising our manuscript (cancers-2599586). We appreciate the reviewer’s thoughtful feedback and constructive comments on this manuscript. We hope the revisions address the reviewer’s comments. You will find the reviewer’s comments with our responses in the following section. Along with our responses, we have provided a clean copy and a highlighted version of the revised manuscript for your convenience. The revision is created in collaboration with all co-authors; each author has approved the final version.

Comments and suggestions for authors:

Minor remarks

  1. Avoid the use of abbreviations in the manuscript. All used abbreviations should be defined in the manuscript. Also, use the defined abbreviation throughout the whole manuscript.

Reply: Thank you for your valuable feedback. We appreciate your suggestion to avoid using abbreviations in the manuscript and to ensure that all abbreviations used are defined within the text. We have made the necessary revisions to comply with this recommendation. We have also used the defined abbreviations consistently throughout the entire manuscript to enhance clarity and readability for the readers; for example, we have replaced glioblastoma with GBM and central nervous system with CNS. Your input contributes to the overall quality of our manuscript.

  1.  

  • The manuscript should be written in the third-person singular. The first-person plural is not acceptable for the scientific manuscript.

Reply: Thank you for your feedback. We acknowledge your suggestion to write the manuscript in the third-person singular and avoid the use of the first-person plural. We requested help from the Language Editing Services at MDPI and have made the necessary adjustments to ensure that the manuscript adheres to this writing style as it is the standard practice for scientific manuscripts. Your input helps to maintain the appropriate level of formality and objectivity in our work.

  1. Many technical errors affect the quality of the manuscript.

Reply: Thank you for your feedback and we appreciate your concern regarding the technical errors in the manuscript. As we take this matter seriously, we have thoroughly reviewed the manuscript to identify and rectify any technical errors that may have impacted its quality. We have improved the data analysis by avoiding certain insignificant figures such as volcano plots and Venn diagrams. Your comments are valuable in our commitment to delivering a high-quality and error-free manuscript. We will ensure that the necessary corrections are made to meet the expected standards.

  1. Each paragraph should be prepared in the same way. For instance, use the same line space, etc. Delete a blank line between two paragraphs.

Reply: Thank you for your observation regarding the formatting of the manuscript. We have carefully reviewed and standardized the formatting of each paragraph to ensure consistency in line spacing and remove any unnecessary blank lines between paragraphs. Your feedback on the manuscript's formatting is appreciated and we will make the required adjustments to enhance its overall uniformity and readability.

  1. Avoid lumping the references. Each reference should be discussed separately.

Reply: Thank you for your feedback, which highlights the importance of individually addressing references rather than lumping them together. We appreciate your suggestion and we will restructure the manuscript to ensure that each reference is discussed separately. We have improved reference lumping; for example, “The fold-enrichment of the x-axis was calculated using -10log (p-value) in the dot plots for the pathways [32, 33, 34–36]” to “The fold-enrichment of the x-axis was calculated using a -10 log (p-value) in the dot plots for the pathways [32], [33], [34]”. This approach enhances the clarity and thoroughness of our discussion of the relevant literature. Your input is invaluable in improving the quality of our manuscript and we will make the necessary revisions.

  1. The figure order should be typed again.

Reply: Thank you for your feedback regarding the figure order in the manuscript. We apologize for any confusion and we have ensured that the figure order is correctly typed and presented. Your attention to detail is greatly appreciated and we have made the necessary adjustments to improve the clarity and organization of the figures in our manuscript, such as Figure 1.

  1. In some figures, the figure labels should be re-scaled to be readable.

Reply: We appreciate your feedback concerning the legibility of figure labels in some of the figures. Ensuring that figure labels are easily readable is crucial for effectively conveying our research. We have carefully reviewed the figures and made the necessary adjustments to re-scale the figure labels as suggested, such as Figure 8B. This enhances the overall quality and comprehensibility of the figures in our manuscript. Your input is valuable and we are committed to addressing this issue to improve the visual presentation of our data.

  1. References should be prepared according to the Instructions for Authors.

Reply: Thank you for your comment regarding the formatting of references. We apologize for any discrepancies and will ensure that all references in the manuscript are prepared according to the specific instructions for authors. We understand the importance of adhering to proper referencing guidelines and have made the necessary adjustments to meet these standards. Your feedback is appreciated and we will work diligently to ensure that the references are correctly formatted in line with the Instructions for Authors.

  1. All minor remarks are depicted in the manuscript.

Reply: Thank you for your acknowledgment and advice on our paper. From the reviewer’s comments, we have rearranged the text and used abbreviations in the whole manuscript. We have written in the third person singular; line spacing within and after paragraphs has been adjusted and is similar throughout the manuscript; the figure order has been re-typed; and specific results have been retained in the main text. We transferred several supporting results to the supplementary figures to ensure clarity of the main text cleaner for readers.

Major remarks

  1. I recommend the reduction of the references list. A literature review of older studies can be omitted from the manuscript. Avoid lumping the references. Each reference should be discussed separately.
  2. Reply: Thank you for your acknowledgment and advice on our paper. We have removed the old studies and discusses the references separately.

  1. The conclusion section is very poor. Please, retype this section and insert all the main conclusions of the research.

Reply: Thank you for your suggestion on our paper. From the reviewer’s comments, we have improved and rewritten the conclusion.

Our results demonstrated that the overexpression of the GNAI3 gene could be an indicator of a poor prognosis and a potential biomarker for GBM. A bioinformatics analysis as well as MetaCore and single-cell analyses revealed that it is involved in the cellular responses that trigger tumor progression and invasion. This marker could be the subject of future in-depth experimental validations.

  We would like to appreciate the editors and reviewers for their interest in this research topic. By addressing all the reviewers’ concerns in a detailed manner, we hope that the improved version of this manuscript will meet the criteria for publication in Cancers. We look forward to your affirmative response.

Sincerely,

Chih-Yang Wang, Ph.D (chihyang@tmu.edu.tw)

Graduate Institute of Cancer Biology and Drug Discovery,

College of Medical Science and Technology,

Taipei Medical University, Taipei, Taiwan.
